# A Review of Document Image Enhancement Based on Document Degradation Problem

**Yanxi Zhou, Shikai Zuo \*, Zhengxian Yang, Jinlong He, Jianwen Shi and Rui Zhang**

School of Opto-Electronic and Communication Engineering, Xiamen University of Technology, Xiamen 361000, China; 2222031329@xmut.edu.cn (Y.Z.)
\* Correspondence: skzuo@xmut.edu.cn

**Abstract:** Document image enhancement methods are often used to improve the accuracy and efficiency of automated document analysis and recognition tasks such as character recognition. These document images could be degraded or damaged for various reasons including aging, fading handwriting, poor lighting conditions, watermarks, etc. In recent years, with the improvement of computer performance and the continuous development of deep learning, many methods have been proposed to enhance the quality of these document images. In this paper, we review six tasks of document degradation, namely, background texture, page smudging, fading, poor lighting conditions, watermarking, and blurring. We summarize the main models for each degradation problem as well as recent work, such as the binarization model that can be used to deal with the degradation of background textures, lettering smudges. When facing the problem of fading, a model for stroke connectivity can be used, while the other three degradation problems are mostly deep learning models. We discuss the current limitations and challenges of each degradation task and introduce the common public datasets and metrics. We identify several promising research directions and opportunities for future research.

**Keywords:** document image enhancement; document image analysis and recognition; deep learning; convolutional neural network; degradation





## 1. Introduction

Document image enhancement is one of the pre-processing steps for document analysis and recognition. The document images are severely damaged or degraded due to long storage time, poor storage environment, and problems such as page smudges and fading of handwriting [1,2].Or, during the digitization process, the degradation of document images is caused by poor lighting conditions, shadows, camera distortion noise, blur, etc. [3–5]. Degraded document images have low visual quality and legibility. They could contain handwritten or machine-printed text or a mixture of both. In addition, they could contain multiple handwriting styles and different languages. To solve this problem, many document image enhancement methods have been proposed. For example, the quality of the image is improved by removing the degradation effect present in the image to restore its original content [6,7]. However, relying only on traditional methods, the task results for these degraded documents still need to be more significant, and it is challenging to automate document analysis. To achieve good results, it is necessary to manually enhance these images, and when the amount of data is relatively large, file image analysis becomes more difficult. Deep learning has been proposed and applied to different computer vision and image processing tasks, such as object detection [8,9], semantic segmentation [10], face detection and dataset creation [11–13], image enhancement [14–16], etc. The data show that compared with traditional methods, methods based on deep learning achieve more gratifying results. Similarly, the method of dealing with document image enhancement has also received great attention. Thus, it is essential to develop methods that can automatically

enhance the visual quality and legibility of these images. The purpose of this survey is to review current methods for degraded document image enhancement and to analyze and discuss the advantages and disadvantages of the leading models for various degradation tasks, focusing on the limitations of the model processing effects and then discussing the directions for future research.

We have structured the rest of the paper as follows. Section 1 summarizes six degradation problems of documents, illustrated in Figure 1. In Section 2, we summarize the models to solve the corresponding problem and review the related work. In Section 3, we present the common public datasets and metrics and discuss the current limitations and challenges of each degradation task. Section 4 summarizes the current data problems in the field of document image enhancement and gives an outlook on the future of the document image enhancement field.

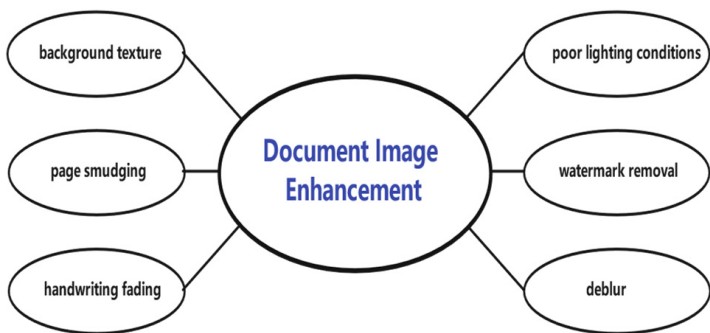

**Figure 1.** Document image enhancement problems.

We make several vital contributions in this paper: one is reviewing the research progress of deep-learning-based document image enhancement methods, especially in the past five years, to help readers better understand the current development in this field. On the other hand, we summarize six image degradation problems, namely, background texture, page smudge, word fading, uneven lighting, watermark, and blur, and analyze the characteristics, advantages, and disadvantages of the leading models to help researchers choose the right approach according to their needs.

## 2. Document Image Degradation Problems

This section describes six degradation problems of document image enhancement, namely, background texture, page smudging, fading, poor lighting conditions, watermarking, and blurring. Figure 2 shows some image examples for each of the degradation problems. The image on the left is the input, and the right image is the output of each problem.

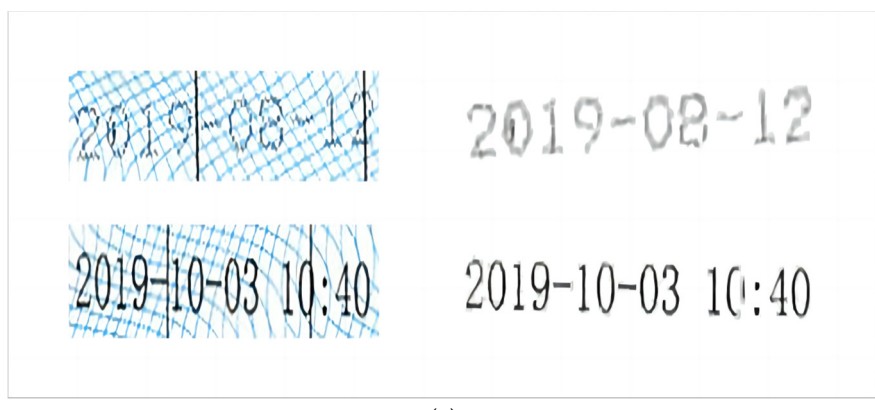

(**a**)

**Figure 2.** *Cont.*

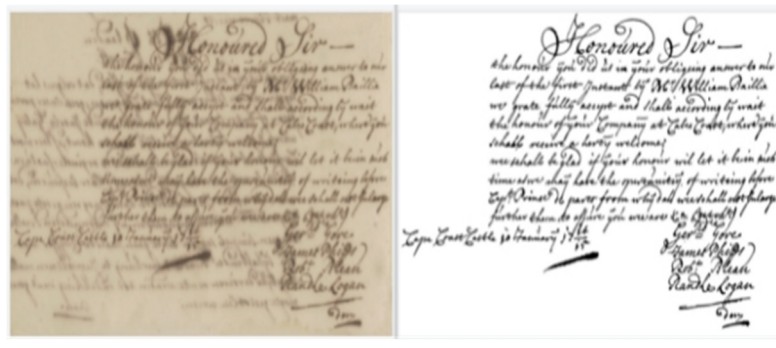

(**b**)

(**c**)

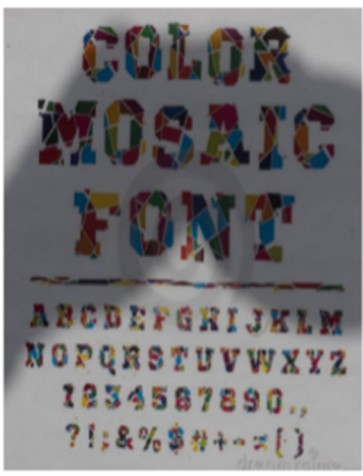
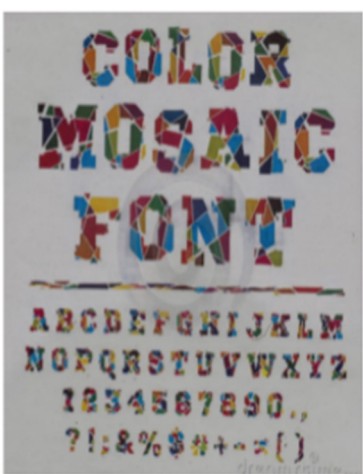

(**d**)

**Figure 2.** *Cont.*

When I was very small, I was so obsessed with the movie Harry Potter.

    I had watched all the series and I was so impressed by the role that acted by Emma Watson. Now so many years have passed, I keep paying attention to her. Emma is as smart as the role she plays. Though she keeps filming movies, she never gives up study. She got full A in her lesson and she could go to any top university, finally she went to Brown University. When she graduated, she was invited to give the speech He For Her. She called the men to be equal with the women. The speech was welcomed by the world and people gave great support. People call Emma the England rose, she is smart and elegant, which sets the good example for the England girls.

**(e)**

**(f)**

**Figure 2.** (**a**) Background texture; (**b**) page smudging; (**c**) handwriting fading; (**d**) poor lighting conditions. Reproduced with permission from Lin et al. [17], CVF Conference on Computer Vision and Pattern Recognition (CVPR); published by IEEE, 2020. (**e**) Watermark removal; (**f**) deblur.

## 2.1. Background Texture

The texture is the description of the spatial distribution pattern of the image pixel gray level, reflecting the texture of the item, such as roughness, smoothness, granularity, randomness, and normalization. However, with the change in document storage time, sometimes the background texture will affect the readability of the text, resulting in the quality of the text being reduced. Figure 2a shows an example of background texture removal.

## 2.2. Page Smudging

Document image files may be incompletely preserved due to damage caused by the natural environment and human factors, resulting in various degradation issues like water impregnation, ink stains, ink mark penetration, and yellowing of pages, among others. These problems can be attributed to the task of handling page smudges in the degradation process. Figure 2b shows an example of addressing page smudges.

## 2.3. Handwriting Fading

The problem of handwriting fading mainly affects historical documents. Ancient historical books are precious pieces of cultural heritage with significant scientific and cultural values. The digitization of ancient books is a necessary approach to document protection and cultural inheritance. However, manual processing of these massive documents is time-consuming and laborious, and error-prone. Therefore, it is necessary to automatically process the images of historical documents with the help of computers. Due to the severe pollution of historical documents, messy writing format, and many other problems, especially the fading of the handwriting caused by these degradation problems, the text recognition of ancient historical documents could be better. Therefore, image analysis and

recognition focusing on historical documents are challenging. Figure 2c is a restoration example of the fading problem.

### 2.4. Poor Lighting Conditions

In the process of document digitization, in the past, scanners were commonly used digital instruments, and scanning resulted in high-quality documents. However, with the development of science and technology, the types of digital image acquisition equipment are increasing, such as digital cameras, mobile phones, cameras, and so on. More people now use their cell phone cameras to get digital copies of their files. However, this approach may encounter problems, such as uneven lighting, excessive shadows, or too dark. In addition, for some old documents or antique literature, there may be wormholes, translucency, or unclear handwriting. This interference with the readability of the acquisition of documents has a harmful impact. The unevenness approach focuses on evaluating shadows on the document image and trying to remove them, making the document clean again and readable. Figure 2d shows an example of shadow removal under uneven lighting.

### 2.5. Watermark Removal

The digital age has led to more accessible communication of digital images and digital information. However, in the process of communication, there are various intentional and unintentional content modifications, and such editing techniques sometimes infringe on intellectual property rights. Moreover, the existence of a watermark is significant for the integrity of original content and copyright protection. Some files may contain one or more watermarks that block the document text, causing problems that make the document difficult to read. Similar to the case of document image denoising, it is difficult to deal with such problems because the watermark color may be the same as or darker than the document text. Therefore, we need a way to recover degraded version files. Watermark removal methods focus on eliminating watermarks and increasing document images' visual quality and readability. We also need to use the correct watermark removal method. Figure 2e shows a watermark-removed image sample.

### 2.6. Deblur

Converting documents to digital images makes it difficult to control some unpredictable degradation factors fully. Among them, how to manage image clarity is an important issue. How to control fuzziness is one of the important concerns. One of the most common examples is the blurring of document images that can be introduced during filming. For example, moving the focus of the lens and camera vibration can all result in blurred images. Figure 2f shows an example of a blurry document image.

Image deblurring technology aims to restore the fuzzy image to the original image. As one of the important research areas in the field of image restoration, there have been many excellent achievements in this regard. The traditional method is mainly the filtering method, generally used to directly restore clear images through deconvolution [18]. With the extensive application of deep learning in recent years, scholars have proposed many effective deblurring models to achieve better results in document image deblurring.

## 3. Document Image Degradation Methods

In this section, we describe document image enhancement methods based primarily on deep learning and discuss their features, challenges, and limitations. Most of these methods solve multiple degradation problems. Below, we describe these methods in more detail.

### 3.1. Background Texture Problem

Methods for processing document images' background texture can be divided into two kinds.

The first method is based on binarization. Image binarization converts the acquired image into a binary image, and the converted image performs better in text recogni-

tion. Binarization can be divided into threshold-based and numerical-based binarization. Threshold-based binarization methods, such as the Otsu algorithm [19], separate the foreground from the background of an image by finding a threshold value that maximizes the variance between the classes. It has the advantage of a fast computation speed for dealing with background texture problems, but the processing effect is still not ideal. Sauvola [20], Niblack [21], and other local threshold algorithms have better separation effects and applicability performance than their global counterparts. However, some problems exist because the thresholding performance depends heavily on the sliding window size. In addition, the parameter tuning method [22], the mixed-threshold algorithm of the clustering method [23], and the advantages of the global threshold and local threshold algorithms are integrated to better segment the foreground and background and reduce the impact of the original image background texture on the document image. Numerical theory-based methods such as the conditional random field (CRF)-based Howe algorithm, [24], which uses the Laplace operator to minimize the target energy function, can have relatively stable results in most cases but has high requirements for image quality. Ref. [25] estimated local threshold values using high-contrast image pixels to binarize the document image.

The second method is based on pixel classification. For example, Bezmaternykh et al. proposed a model [26] based on U-Net. They combined it with the structural advantages of CNN for historical document images to improve the ability of the model to obtain context information. However, manual parameter adjustment and data enhancement models are needed, and the model's generalization ability needs to be improved. In 2021, Xiong et al. proposed an FD-Net model [27], as shown in Figure 3. It is a full dilation convolutional network, replacing all the downsampled and upsampled layers with dilation convolution. It can enhance the receptive field and segmentation accuracy. However, a grid effect, i.e., a discontinuity in the convolution kernel, may arise, which is unfavorable for the pixel-by-pixel prediction task.

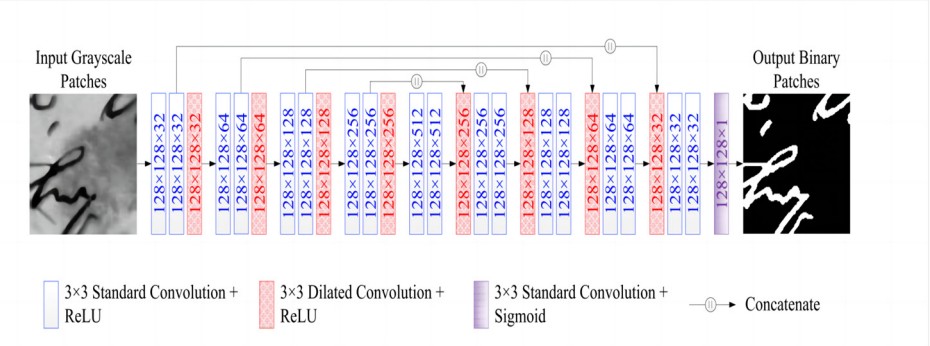

**Figure 3.** The structure of FD-Net, constructed by Xiong et al. [27]. Reproduced with permission from Xiong et al., Springer eBook; published by Springer Nature, 2021.

*3.2. Page Smudging Problem*

Problems include ink penetration, prominent smudges, and yellowing of paper in the original document due to improper preservation. We attribute this type of problems to page smudging in degradation tasks. In addition to the traditional binarization method, the following models have better effects on this kind of problem.

In 2018, a hierarchical deep supervised network (DSN) architecture [28] was proposed by Vo et al. It applied the deep supervised network model to the document image binarization issues for the first time, which could better distinguish the foreground from background noise to predict text pixels with different feature levels. However, the large number of convolutional layers leads to the problem of processing small amounts of data as well as long processing times. In the same year, Bhowmik et al. proposed a game theory-inspired binarization technique [29], which designs a two-player, non-zero-sum algorithm at the pixel level to extract local information from document images, and then feeds the

obtained results into the K-means algorithm for pixel classification. The pre-processing step and the post-processing step of the method also contribute to the enhancement of the overall algorithm, eliminating the intensity variations that often occur in the background and refining the results. In 2019, Gallego et al. proposed a selective auto-coding approach [30] that performs a transformation step on the input signal using a convolutional auto-encoder and adjusts these transformation parameters during the training process to achieve model optimization, and finally applies a global threshold to binarize the document, as shown in Figure 4. For different input image sizes and when the convolutional network uses a different number of convolutional kernels, the model can have good binarization results, and the convolutional kernel size does not have a large impact on the binarization results. However, the number of model parameters needs to be increased due to the use of step size instead of the pooling layer in the downsampling process. In the same year, Zhao et al. introduced conditional generative adversarial networks (cGAN) to solve the core problem of multi-scale information combinations in binarization tasks [31], as shown in Figure 5. The model has a good effect on degradation problems such as page smudging, but the drawback is that the training strategy of the sub-generator still needs to be modified, and the update efficiency and scalability of the model still need to be improved. Peng et al. used a multi-resolution attention model to infer the probability of text areas and then feed it into a convolutional random field (ConvCRF) to obtain the final result [32]. This method uses distance reciprocal distortion measure loss to infer the relationship between text regions and background by ConvCRF, which has a stronger generalization ability. However, the training and inference speed of ConvCRF is slow compared to other models, and its parameters are difficult to learn.

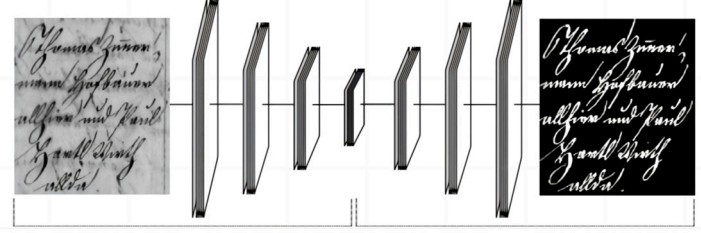

**Figure 4.** Model structure constructed by Jorge and Gallego [30]. The output layer consists of the activation level assigned to each input feature (white denotes maximum activation). Published under a Creative Commons Attribution 4.0 license.

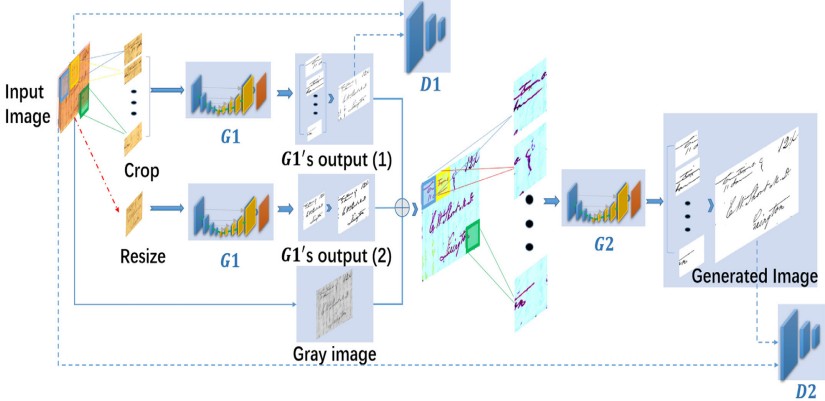

**Figure 5.** The structure of the model constructed by Zhao et al. [31]. The whole-image generator is composed of G1 and G2. The two sub-generators are cascaded, and G2 optimizes the results of G1. Reproduced with permission from Zhao et al., *Pattern Recognition*; published by Elsevier, 2019.

In 2020, Souibgui et al. proposed the model of DE-GAN [33], which transforms the threshold setting task of processing text and background into an image-to-image

transformation task. For the generator, a U-Net-based structure is used, and the addition of bypass links reduces information loss and allows the model to learn more advanced features. For the discriminator, a simple multilayer convolution structure is used. The model cannot only deal with degradation problems such as page smudging but also has better results in watermark and blur removal for document images. Although it has better results on different document enhancement tasks, when dealing with binarization tasks, it is more time-consuming to train it several times in processing the problem to obtain better results because of the need to manually change the variables in the loss function.

*3.3. Handwriting Fading Problem*

In digitizing many historical books, the document images have a degraded condition of faded handwriting. A series of models have been proposed for this situation in image processing.

In 2018, Jia et al. proposed an adaptive global threshold selection algorithm based on structural symmetric pixels (SSP) [34]. First, they proposed an adaptive global threshold selection algorithm for SSP extraction, which can remove bleeding backgrounds while preserving fuzzy characters. SSP extraction enables feature maps of arbitrary size to output a vector of fixed length. Then, the proposed stroke width estimation algorithm improves adaptiveness. Finally, an iterative algorithm is used to estimate the text stroke width and remove the noise at the same time. This method is more suitable for processing printed text images than handwritten text images. In 2019, Ref. [35] introduced an adversarial learning approach, as shown in Figure 6. This method constructs a Texture Augmentation Network that transfers the texture element of a degraded reference document image to a clean binary image. The available document dataset is expanded by adding various noisy textures to the same textual content to form multiple versions of the degraded image. Finally, the newly generated images are passed through the network in order to return clean versions. The model is effective in improving robustness, and it does not require paired data. The model is similar to the cycleGAN approach, but the results obtained are more advantageous, thanks to the ability of BiNet in the model to learn autonomously and to fully supervise the optimization results.

In 2021, Xiong et al. proposed an algorithm based on background estimation and energy minimization [36]. First, a disc-shaped structuring element whose radius is computed by the minimum entropy-based stroke width transform (SWT) is used to estimate and compensate for the document background. SWT can extract the stroke width to obtain the character candidate region and filter out the mis-checked candidates using a priori knowledge of shape and texture. Then, Laplacian energy-based segmentation is performed on the compensated document images. Finally, post-processing is implemented to preserve text stroke connectivity and eliminate isolated noise. This method is able to better preserve text strokes, but it does not have more advantages than the current advanced deep-learning-based methods. In the same year, the DP-LinkNet model [37] was proposed by Xiong et al. The model follows the LinkNet [38] and D-LinkNet [39] architectures with the addition of the hybrid dilated convolution (HDC) module and the spatial pyramid pooling (SPP) module, which expands the perceptual field and aggregates multiscale contextual features, and an SPP module that encodes the output of the HDC with multicore pooling, both of which enable the model to extract textual strokes features with deep semantic information. Compared to other neural network models, training is faster and does not require deeper network layers. In 2022, Suh et al. proposed a two-stage color document image enhancement and binarization method using generative adversarial neural networks [40]. The first stage uses a color-independent adversarial neural network to remove background information from local image blocks and extract colored foreground information for document image enhancement. The second stage uses a multiscale adversarial neural network to generate a local binarized resultant image and a global binarized resultant image of the document image. One of the model's advantages is its ability to handle multi-color degradation of document images. The disadvantage is that the training is slow due to the

need to generate images of different channels. In the same year, Souibgui et al. proposed an encoder–decoder architecture based on Vision Transformer [41], as shown in Figure 7. The degraded image is first divided into several patches, which are then fed into the encoder, where the patch is mapped to a potential representation of each token during the encoding process, where each token corresponds one-to-one. The token is then passed to the decoder for patch enhancement. The model uses a self-attentive mechanism to obtain high-level global information for better performance. Since the images are trained in pixel blocks, there may be incomplete extraction of information from edge junction regions.

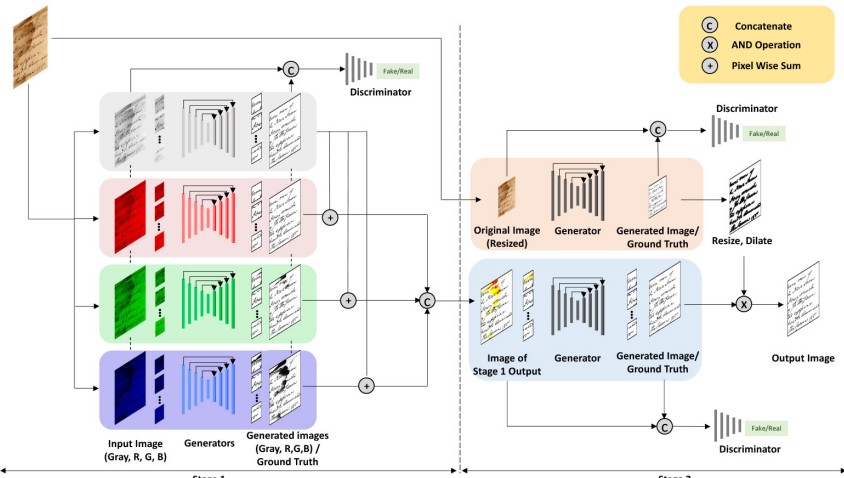

**Figure 6.** The structure of the model constructed by Suh et al. [40]. Reproduced with permission from Suh et al., *Pattern Recognition*; published by Elsevier, 2022.

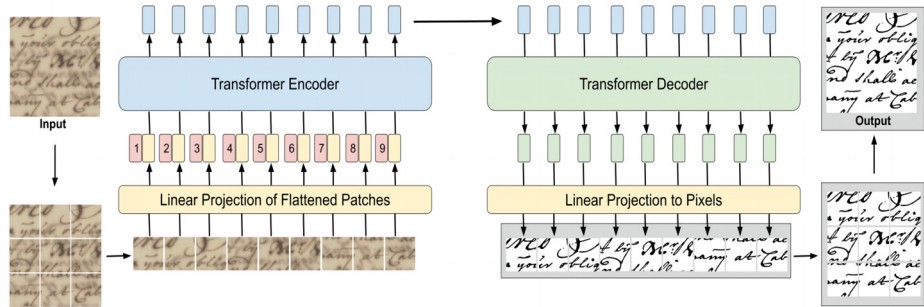

**Figure 7.** DocEnTr model structure constructed by Souibgui et al. [41]. The input image is split into patches, which are linearly embedded, and the position information is added to them. The resulting sequence of vectors is fed to a standard Transformer encoder to obtain the latent representations. These representations are fed to another Transformer representing the decoder to obtain the decoded vector, which is linearly projected to vectors of pixels representing the output image patches. Published under a Creative Commons Attribution 4.0 license.

### 3.4. The Problem of Poor Lighting Conditions

Due to the effects of uneven lighting, various shadows are produced in the captured images. The following is a proposed method for removing shadows to address the problem of uneven lighting.

In 2017, Bako et al. proposed a method to estimate the background and text colors in local image blocks [42]. First, the local background color estimates are matched to the global reference to generate a shadow map. This shadow map is then used to correct the image to produce a final unshadowed output. This method detects and removes shadows from document photos and other text-based items. However, there are some limitations; e.g., the result will produce slight color and brightness variations, or in cases with strong hard

shadows, the method can leave small residual artifacts at the shadow boundaries. In 2018, Ref. [43] proposed the ST-CGAN model, which consists of two stacked CGANs, each with a generator and a discriminator, capable of performing two tasks, shadow detection and shadow removal. It is the first framework for end-to-end joint learning of shadow detection and shadow removal with superior performance on various datasets and on both tasks. However, it is more demanding on the dataset and has more a priori conditions. In the same year, Kligler et al. introduced a newer general approach to the document enhancement problem [4], the main idea being to modify any state-of-the-art algorithm to improve its results by providing it with new information (input). Finally, it was found that the algorithm can be used for visibility detection and has better results for document shadow removal. However, for slight shadow problems, the PSNR effect of the output image does not have an advantage. In 2019, Jung et al. proposed an efficient shadow removal algorithm for digitized documents [44], which uses the luminance value of each pixel to create a digitized input document and then estimates the shading artifact on the document by simulating the immersion process before correcting the illumination. After estimating the shading artifacts, the digitized document is reconstructed using the Lambertian surface model. However, for the task of eliminating specularly reflected light from the image, the model does not process very well. In 2020, Lin et al. proposed the BEDSR-Net model [17], the first work to propose using deep learning to remove document shadows, as shown in Figure 8 below. The authors designed a background estimation module to extract the global background color of a document, as well as to learn information about the spatial distribution of pixels, encode it into an attention map, and then use a shadow removal network (SR-Net) to complete the shadow removal work. However, the effect of removing images with low background brightness needs to be improved.

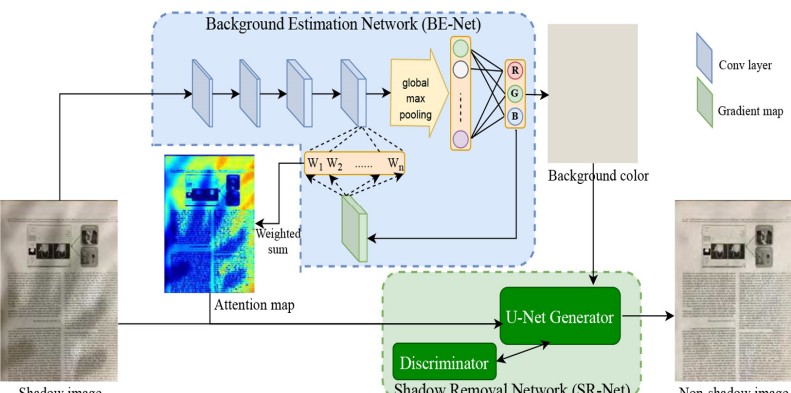

**Figure 8.** BEDSR-Net model structure constructed by Lin et al. [17]. It consists of two sub-networks: BE-Net for estimating the global background color of the document and SR-Net for removing shadows. Given the input shadow image, BE-Net predicts the background color. As a side product, it generates an attention map, which depicts how likely it is that each pixel belongs to the shadow-free background. Along with the input shadow image, the estimated background color and the attention map are fed into the SR-Net for determining the shadow-free version of the input shadow image. Reproduced with permission from Lin et al., CVF Conference on Computer Vision and Pattern Recognition (CVPR); published by IEEE, 2020.

In 2021, CANet [45] was proposed by Chen et al., as shown in Figure 9. The innovation of this model is to achieve shadow removal by passing information from non-shadowed regions to shadowed regions in the feature space, but the poor precision and inconsistent color of the segmented blocks limit it. Ref. [46] proposed the LG-ShadowNet model for shadow removal by training on unpaired data, where shaded and unshadowed training images are completely different and have no correspondence. The model does not use particularly sophisticated methods to optimize the results but rather uses linear regression

to transform the pixel values to achieve better results and reduce errors. However, the advantages are not very obvious in terms of training for paired data.

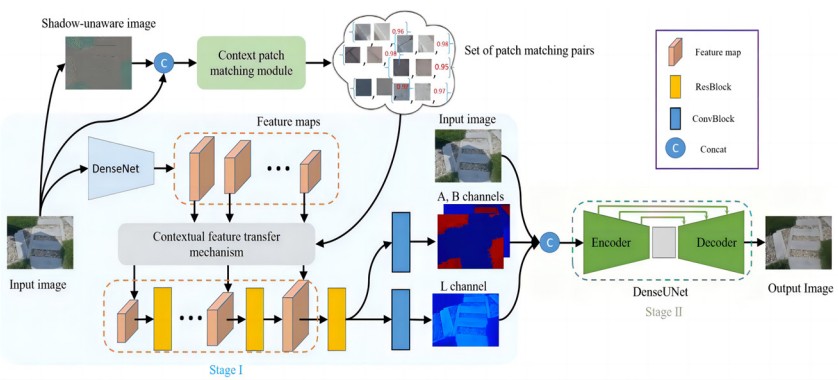

**Figure 9.** CANet model structure constructed by Chen et al. [45]. At Stage I, the contextual feature is firstly extracted via a pre-trained DenseNet; meanwhile, the designed contextual patch matching module (CPM) is used to acquire a set of contextual matching pairs; then, a contextual feature transfer mechanism is applied to transfer contextual information from non-shadow patches to shadow patches to recover the L and A/B channels of the shadow-removal image. At Stage II, the recovered L and A/B channel information is integrated with the input shadow image and fed into a DenseUNet to generate the final shadow-removal result. Published under a Creative Commons Attribution 4.0 license.

### 3.5. Watermark Removal Problem

Watermarking is the most common way to protect the copyright of an image. Many organizations will add watermarks to their product images to prevent them from being stolen by others. We must recover the real image from the watermarked image if the original image is lost. Watermark removal can be divided into two main methods: the first is a single-stage target detection method, and the second is a two-stage target detection method. Next, typical models of the two methods are listed.

The single-stage target detection method is as follows.

In 2019, Gangeh et al. designed and tested an encoder–decoder model with an eight-level jump connection [47] to improve the quality of scanned documents with better results for removing watermarks from documents, but experimental and quantitative evaluations were inadequate. In 2020, DE-GAN was proposed [33], a more flexible model that is useful for watermark removal and other document degradation problems such as deblurring. However, it is computationally complex and requires predetermined parameters to be adjusted for different images. In 2022, Ref. [48] proposed an end-to-end architecture based on GANs with the addition of a handwritten text recognizer, where text information patches exist in handwritten text recognition (HTR) form during training to increase the readability of the text and facilitate document watermark removal, as shown in Figure 10. However, the use of the results for unpaired data is not yet as advantageous.

The two-stage target detection method is as follows.

In 2019, Liu et al. proposed a novel two-stage separable deep learning (TSDL) framework [49] consisting of noise-free end-to-end adversary training (FEAT) and noise-aware decoder-only training (ADOT). A redundant multilayer feature encoding network (RM-FEN) was designed as an encoder framework that can learn a robust watermarking pattern without seeing any noise, can resist high-intensity noise, and has better stability, greater performance, and faster convergence speed. However, it is necessary to set the parameters manually, resulting in the need to manually set the results of the algorithm according to the features of different images, which leads to a decrease in model efficiency when facing multiple datasets. In 2020, Jiang et al. proposed a visible watermark removal network architecture [50] based on conditional generative adversarial networks (CGAN) [51] and least squares generative adversarial networks (LSGAN) [52]. As shown in Figure 11, the

proposed model can automatically detect the watermark location without manual checking, which improves the model efficiency, but the generalization ability of the model is weak.

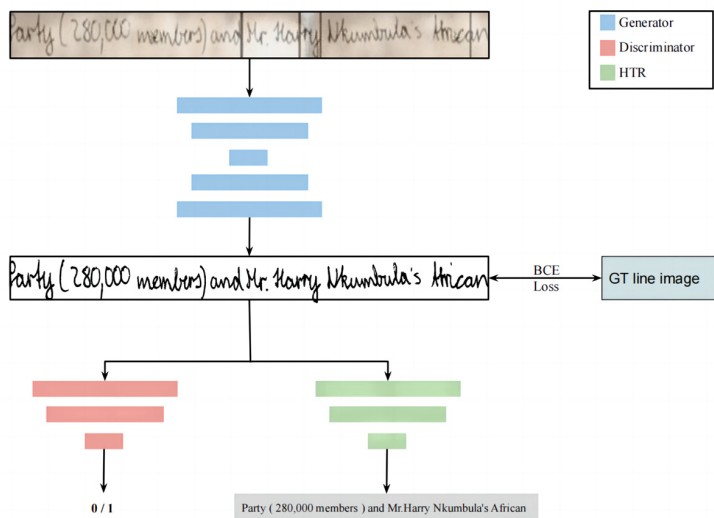

**Figure 10.** The structure of the model constructed by Jemni et al. [48]. Published under a Creative Commons Attribution 4.0 license.

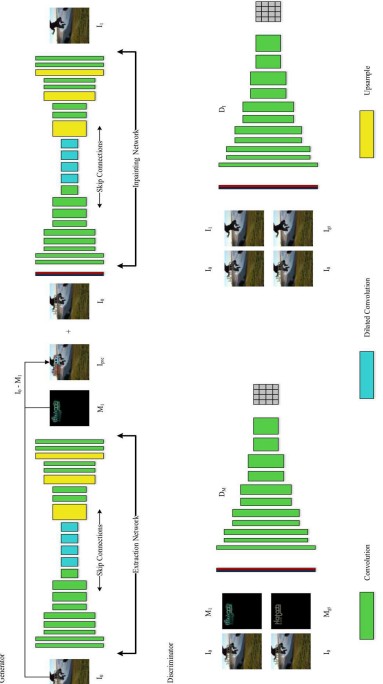

**Figure 11.** Two-stage visible watermark removal architecture constructed by Jiang et al. [50]. Reproduced with permission from Jiang et al., *IET Image Processing*; published by John Wiley and Sons, 2021.

In 2021, WDNet was proposed by Liu et al. [53], which is a network dedicated to watermark removal. It does not remove the watermark but separates the watermark and uses this watermark to build a larger dataset, which can better segment the real image and the watermark with higher accuracy. However, the segmentation accuracy for color images still needs to be improved. In 2023, Ge et al. proposed an end-to-end document image watermarking scheme using deep neural networks [54], as shown in Figure 12. A noise layer is added to simulate various attacks that may be encountered in reality, such as cropout, dropout, Gaussian blur, etc. The text-sensitive loss function aims to limit the

embedding modification of characters. An embedding strength adjustment strategy is also proposed in the paper to improve the quality of watermarked images while reducing the loss of extraction accuracy. This method is designed specifically for document images, but the advantages of processing the natural image watermarking scheme are not great.

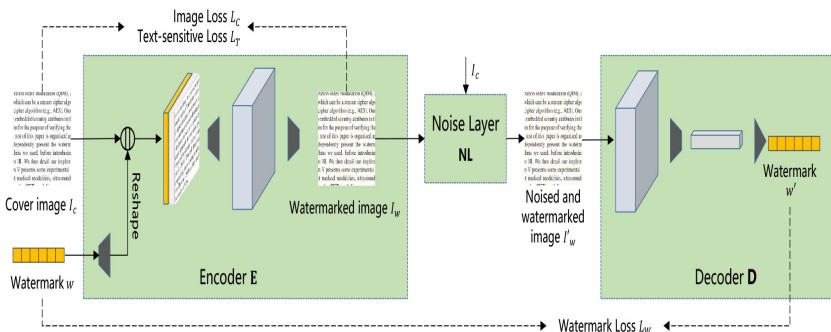

**Figure 12.** Structure of the model proposed in [54], constructed by Ge et al. In the training stage, the watermark w is expanded, reshaped, and concatenated with the cover image $I_c$. Then, a combination of the cover image $I_c$ and the watermark w is encoded to generate the watermarked image $I_w$. Next, $I_w$ is input into the noise layer which simulates the attacks on input, generating the noised and watermarked image $I'_w$. Finally, $I'_w$ is input into Decoder D to extract the watermark $w'$ which could be similar to or the same as w. After training, Encoder E is used to embed the watermark and Decoder D is used for watermark extraction. Reproduced with permission from Ge et al., *Multimedia Tools and Applications*; published by Springer Nature, 2023.

*3.6. Deblur Problem*

The process of acquiring document images (e.g., scanned images of archival contracts) in the batch may be subject to various disturbances, the most common of which are motion-type blurs such as camera shake and motion offset. Document image blurring not only makes the image content visually illegible but also leads to poor image text detection and recognition, so in-depth research on document image deblurring is needed.

In 2015, Hradis et al. proposed using convolutional neural networks to recover blurred text documents [55]. The trained network can reconstruct high-quality images directly from blurred inputs without assuming any specific blur and noise model. The performance of convolutional networks was improved on a large number of text documents and a combination of realistic bokeh and camera-dithering blur kernels. This method gives good results even if the noise level is higher than the trained model. However, the model still has room for improvement. In 2016, Pan et al. proposed a simple and effective blind image deblurring method based on the dark channel prior [56]. Based on the observation of the blurring process, the L0 regularization term is used to minimize the dark channel. In the recovery process, it is in favor of the clear image rather than the blurred image to improve the model's deblurring performance. The model also has good results in non-uniform deblurring, but the accuracy of the processing effect on text images needs to be improved. In 2017, a model for cyclic consistency was proposed in Ref. [57]. It is a method for learning to transform images from a source domain X to a target domain Y without paired examples and is able to capture the relationship between high-level appearance structures. However, it has drawbacks; e.g., the accuracy is not high enough. In the same year, Nah et al. applied deep multiscale convolutional neural networks to the problem of deblurring [58]. This multiscale strategy decomposes the complex problem and recovers it step by step, first recovering large-scale information at low resolution and then recovering detailed information at high resolution. The problem is simplified while increasing the receptive field of the image as a way to improve the model's ability to obtain context. However, the deblurring effect on real blurred images needs to be improved. In 2018, DeblurGAN is a GAN-based method for blind motion blur removal proposed by Kupyn et al. [59], who were inspired by SRGAN [60] and CGAN [51] and regarded image deblurring as a

special class of image2image task, which is based on WGAN [61] and content loss to train the model with good performance in terms of processing speed and results. The model is mainly for the dataset with high-speed motion, and the effect for ordinary fuzzy data needs to be improved. In 2019, Lee et al. came out with a blind text image deblurring algorithm using a text-specific hybrid dictionary [62]. It is a text image prior based on sparse representation, which models the relationship between an intermediate latent image and a desired sharp image and can obtain better contextual information. However, it is not effective if the blurred text image has low contrast. In the same year, Lu et al. proposed an unsupervised method for domain-specific single-image deblurring based mainly on disentangled representation [63] by untangling content and blurred features in blurred images and adding KL divergence loss to prevent blurred features from encoding content information. To preserve the content structure of the original image, a blurring branch and cyclic-consistency loss are added to the framework, while the added perceptual loss helps to remove unrealistic artifacts from the blurred image. The innovation is precisely the attempt to separate content from ambiguous information. In 2020, a method using the Local Maximum Difference Prior (LMD) was proposed in Ref. [64] by introducing a linear operator to compute the LMD and using the L1 norm to constrain the terms involved in the LMD. The LMD method decomposes a complex multicomponent signal into several components. This method is more general. However, when the image contains non-Gaussian noise, the processing is not very effective. In 2021, Blur2sharp was proposed by Neji et al. as an end-to-end model based on cycleGan [65], as shown in Figure 13. This model can perform deblurring without knowledge of the blurring kernel and generates clear images. However, the model processing results are similar to those of other methods and are not particularly advantageous. In 2022, Gonwirat et al. proposed DeblurGAN-CNN [66], as shown in Figure 14, a combination of DeblurGAN and CNN models which was upgraded from the effect of the original DeblurGAN to solve many noise problems and enhance the recognition performance of handwritten character images.

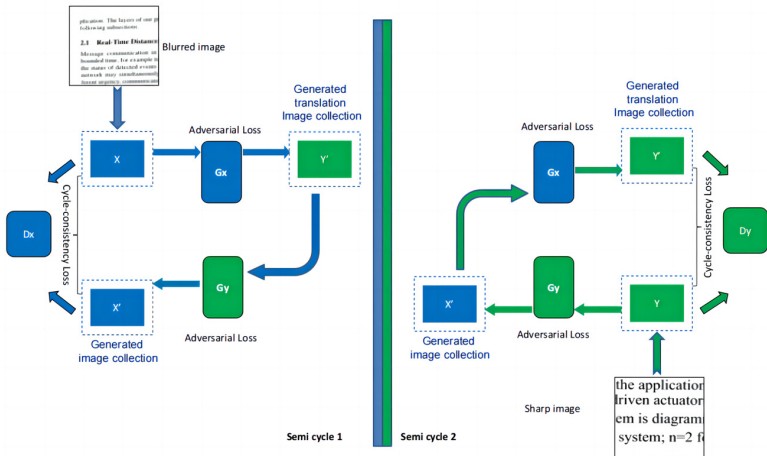

**Figure 13.** Training process of blur2sharp model structure constructed by Neji et al. [65]. Published under a Creative Commons Attribution 4.0 license.

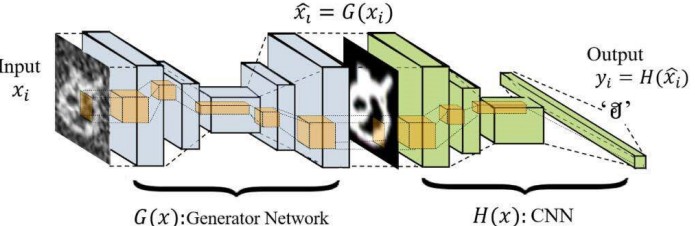

**Figure 14.** DeblurGAN-CNN model structure constructed by Gonwirat and Sarayut [66]. Published under a Creative Commons Attribution 4.0 license.

## 4. Datasets and Metrics

*4.1. Datasets*

1.  Jung's dataset [44]

    The dataset consists mainly of 159 digitized documents with illumination distortion, of which 109 images were captured using the cameras of two smartphones and the remaining 50 were captured using a scanner. The size of the camera-captured images was $3264 \times 2448$ and that of the scanned images was $3455 \times 2464$ (72 dpi).

2.  Real Document Shadow Removal Dataset(RDSRD) [17]

    The images were captured using Sony RX100 m3 and a flashlight, manufactured by the Sony group of companies from Tokyo, Japan. These devices are all fixed on tripods The camera is triggered using a remote through WiFi to avoid touching the camera during capture. The dataset contains 540 photos, including images of the effects of paper, newspapers, and slides under different lighting conditions, covering 25 documents.

3.  Blurry document images (BMVC)Text dataset [55]

    A total of 66,000 text images of size $300 \times 300$ were used for training and 94 images of size $512 \times 512$ were used for OCR testing. Each patch was extracted from a different document page and each fuzzy kernel used was unique.

4.  Bickley diary [67]

    The images of the Bickley diary dataset were taken from a photocopy of a diary that was written about 100 years ago. It consists of 92 badly degraded handwritten travel diary documents, 7 of which have GT images. These images suffer from different types of degradation, such as water stains and ink bleed-through.

5.  SMADI (Synchromedia Multispectral Ancient Document Images Dataset) [68]

    S-MS (Synchromedia Multispectral Ancient Documents): Multispectral imaging (MSI) represents an innovative and non-destructive technique for analyzing materials such as ancient documents. Ancient handwritten letters were collected in a multispectral image database. This database contains 30 multispectral images of authentic historical handwritten letters. These old documents are written in iron gall ink and are dated from the 17th to the 20th century. The original documents were borrowed from the National Library of Quebec and were imaged using an eighth-note CX MSI camera. Through this process, 8 images were produced for each document, resulting in a total of 240 images for the file.

6.  DIBCO and H-DIBCO

    The DIBCO printed or handwritten document image dataset is mainly used for the task of binarization. It was introduced by the Document Image Binarization Competition, including DIBCO 2009 [69], 2011 [70], 2013 [71], 2017 [72], and 2019 [73] and H-DIBCO 2010 [74], 2012 [75], 2014 [76], 2016 [77], and 2018 [78] benchmark datasets, covering 36 machine-printed and 90 handwritten document images, as well as their corresponding ground truth images. The historical documents in these datasets originated from the Recognition and Enrichment of Archival Documents (READ) project. A total of 136 test images contain representative historical document degradation, such as severely damaged pages, ink bleed-through, page stains, text stroke fading, and background texture.

7.  DocImgEN

    A batch of PDFs were downloaded from the IEEE Xplore database [79] and converted PDF pages to PNG images. Then, the image block size of $400 \times 400$ was cropped to the document watermark image. DocImgEN includes 230,000 training samples, 10,000 validation samples, and 10,000 test document images with English words.

8. DocImgCN

A batch of PDFs were downloaded from the China National Knowledge Infrastructure (CNKI) [80] to prepare document images with Chinese characters. DocImgCN includes 230,000 training samples, 10,000 validation samples, and 10,000 test document images.

Table 1 shows a summary of the relevant contents of the dataset.

**Table 1.** Degradation problems targeted by the datasets.

| Dataset | Task | No. of Image | Real vs. Synthetic |
|---|---|---|---|
| Jung's dataset | poor lighting conditions | 159 | real |
| RDSRD | poor lighting conditions | 540 | real |
| BMVC | deblurring | 3M train/35K test | synthetic |
| Bickley diary | multiple | 7 | real |
| SMADI | multiple | 240 | synthetic |
| DIBCO and H-DIBCO | multiple | 10, 10 | real |
| DocImgEN | watermark removal | 10 k train/10 k validation/10 k test | synthetic |
| DocImgCN | watermark removal | 230 k train/10 k validation/10 k test | synthetic |

*4.2. Metrics*

In this section, for the quantitative evaluation and comparison of advanced algorithms, we introduce evaluation metrics: F-measure (FM), inverse distance distortion (DRD), pseudo-F-measure (p-FM), peak signal-to-noise ratio (PSNR), word error rate (CER), word error rate (WER), and structural similarity (SSIM).

- F-Measure (FM) [71]: It is the summed average of accuracy and recall, which is a common evaluation criterion in the field of IR (information retrieval) and is often used to evaluate the classification models. The calculation formula is as follows:

$$\mathbf{F_\beta} = \frac{\left(\beta^2 + 1\right)\mathbf{PR}}{\beta^2 \cdot \mathbf{P} + \mathbf{R}} \tag{1}$$

where $\beta$ is a parameter, P is precision, and R is recall. $\mathbf{P} = \frac{\mathbf{TP}}{\mathbf{TP+FP}}$, $R = \frac{\mathbf{TP}}{\mathbf{TP+FN}}$, TP denotes true, FP denotes false positive, and FN denotes false negative.

When the parameter $\beta = 1$, it is the most common F1-measure.

$$F_1 = \frac{2 \times PR}{P + R} \tag{2}$$

- Distance Reciprocal Distortion (DRD) [71]: It is used to measure the visual distortion of the image in the binary document. The formula is as follows:

$$DRD = \frac{\sum_k DRD_k}{NUBN} \tag{3}$$

where $DRD_k$ is the distortion of the kth flipped pixel and NUBN is the number of non-uniform (not all black or white pixels) $8 \times 8$ blocks in the GT image.

- Pseudo-F-Measure: It is introduced in Ref. [81], which makes use of pseudo-recall and pseudo-precision. The advantage of using pseudo-recall and pseudo-precision is that they use the weighted distance between output images as boundaries of characters in the extracted document and the boundaries of characters in the ground truth (GT) image. One other advantage of using the pseudo nature of recall is its consideration of local stroke width in output images, while pseudo-recall takes into consideration the

local stroke width, and the pseudo nature of precision grows to the stroke width of the connected component in ground truth images.

$$p - FM = \frac{2 \times p\,\text{Recall} \times \text{Precision}}{p\,\text{Recall} + \text{Precision}} \tag{4}$$

where *p*Recall is defined as the percentage of the skeletonized ground truth image.

- Peak signal-to-noise ratio (PSNR) [82]: The ratio between the maximum possible power used to represent a signal and the power of the corrupted noise that affects the fidelity of its representation. Since many signals have a wide dynamic range, PSNR is often expressed as a pair number using a decibel scale. PSNR is mainly used in image processing to quantify the reconstruction quality of images and videos affected by lossy compression.

PSNR is defined by the mean square error (MSE). Given a noise-free m × n monochrome image I and its noise approximation K, the MSE is defined as

$$\text{MSE} = \frac{1}{mn}\sum_{i=0}^{m-1}\sum_{j=0}^{n-1}[I(i,j) - K(i,j)]^2 \tag{5}$$

PSNR is defined as

$$\text{PSNR} = 10 \cdot \log_{10}\left(\frac{MAX_I^2}{\text{MSE}}\right) \tag{6}$$

where $MAX_I^2$ is the maximum possible pixel value of the image.

- Character error rate (CER) [83]: Character Error Rate is computed based on the Levenshtein distance. It is the minimum number of character-level operations required to transform the ground truth or reference text into the OCR output text. CER is formulated as follows:

$$\text{CER} = \frac{S + D + I}{N} \tag{7}$$

where S is the number of substitutions, D is the number of deletions, I is the number of insertions, and N is the number of characters quoted or ground truth text. The lower the CER value, the better the OCR performance model. CER can be normalized to ensure that it does not fall out of the range of 0–100 and is not affected by insertion errors. In normalizing CER, C is the number of correctly identified in the text sentence. The normalized CER is formulated as follows:

$$\text{CER}_{\text{normalized}} = \frac{S + D + I}{S + D + I + C} \tag{8}$$

- Word error rate (WER) [84]: In order to maintain consistency between the identified word sequences and the standard word sequences, certain words need to be replaced, deleted, or inserted, and the total number of these inserted, replaced, or deleted words, divided by the percentage of the total number of words in the standard word sequences, is the WER.

$$\text{WER} = \frac{S_w + D_w + I_w}{N} \tag{9}$$

where $S_w$ is the number of substitutions, $D_w$ is the words deleted, $I_w$ is the words inserted, and N is the number of words.

- Structural similarity (SSIM) [85]: The structural similarity index defines structural information from the perspective of image composition as an attribute that reflects the structure of objects in a scene independently of luminance and contrast and models distortion as a combination of three different factors: luminance, contrast, and

structure. Given patch "x" from one image and corresponding patch "y" from another image. The formula is as follows:

$$\text{SSIM}(\boldsymbol{x}, \boldsymbol{y}) = \frac{\left(2\mu_x\mu_y + C_1\right)\left(2\sigma_{xy} + C_2\right)}{\left(\mu_x^2 + \mu_y^2 + C_1\right)\left(\sigma_x^2 + \sigma_y^2 + C_2\right)} \tag{10}$$

where $\mu_x$ is the average of $\boldsymbol{x}$, $\mu_y$ is the average of $\boldsymbol{y}$, $\sigma_x^2$ is the variance of $\boldsymbol{x}$, $\sigma_y^2$ is the variance of $\boldsymbol{y}$, and $\sigma_{xy}$ is the covariance of $\boldsymbol{x}$ and $\boldsymbol{y}$. $C_1$, $C_2$ stand for constant.

### 4.3. Experiment

We conducted experiments based on 10 DIBCO and H-DIBCO test datasets for the background texture and page smudge problems. The handwriting fading problem was investigated with experiments for the DIBCO2018 dataset. The results are shown in Table 2. The experimental results of the illumination non-uniformity problem are shown in Table 3. The experimental results of the deblurring problem are shown in Table 4.

**Table 2.** Experimental results of background texture, page smudging, and fading of lettering.

| Degradation | Dataset | Method | FM | $F_{PS}$ | PSNR | DRD |
|---|---|---|---|---|---|---|
| background texture | | Otsu | 74.22 | 76.99 | 14.54 | 30.36 |
| | | Niblack | 41.12 | 41.57 | 6.67 | 91.23 |
| | | Sauvola | 79.12 | 82.95 | 16.07 | 8.61 |
| | | Bezmaternykh's UNet | 89.29 | 90.53 | 21.32 | 3.29 |
| | DIBCO | FD-Net | 95.25 | 96.65 | 22.84 | 1.22 |
| | H-DIBCO | Vo's DSN | 88.04 | 90.81 | 18.94 | 4.47 |
| | | Bhowmik's GiB | 83.16 | 87.72 | 16.72 | 8.82 |
| page smudge | | Gallego's SAE | 79.22 | 81.12 | 16.09 | 9.75 |
| | | Zhao's cGAN | 87.45 | 88.87 | 18.81 | 5.56 |
| | | Peng's woConvCRF | 86.09 | 87.40 | 18.99 | 4.83 |
| | | Bhunia [35] | 59.25 | 59.18 | 11.80 | 9.56 |
| | | Xiong [36] | 88.34 | 90.37 | 19.11 | 4.93 |
| handwriting fading | H-DIBCO 2018 | DP-LinkNet | 95.99 | 96.85 | 22.71 | 1.09 |
| | | Suh [40] | 84.95 | 91.58 | 17.04 | 16.86 |
| | | DocEnTr | 90.59 | 93.97 | 19.46 | 3.35 |

**Table 3.** Experimental results of uneven illumination.

| Degradation | Method | Jung's Datasets | | RDSRD | |
|---|---|---|---|---|---|
| | | PSNR | SSIM | PSNR | SSIM |
| | Bako [42] | 23.70 | 0.9015 | 28.24 | 0.8664 |
| | ST-CGAN | 23.71 | 0.9046 | 30.31 | 0.9016 |
| poor lighting conditions | Kligler [4] | 24.45 | 0.8332 | 22.53 | 0.7056 |
| | Jung [44] | 28.49 | 0.9108 | 14.45 | 0.7054 |
| | BEDSR-Net | 27.23 | 0.9115 | 33.48 | 0.9084 |

**Table 4.** Defuzzification experimental results.

| Degradation | Dataset | Method | PSNR | SSIM | CER |
|---|---|---|---|---|---|
| | | Hradis [55] | 30.6 | 0.98 | 7.2 |
| | | Pan [56] | 21.84 | 0.93 | 35.3 |
| deblurring | BMVC text | Zhu [57] | 19.57 | 0.89 | 53.0 |
| | | Nah [58] | 22.27 | 0.92 | 50.6 |
| | | Lu [63] | 22.56 | 0.95 | 10.1 |

For the background texture problem, it can be seen from the experimental results that the deep learning approach is more advantageous compared to the traditional thresholding approach; in particular, the FD-Net model effect is excellent in the comparison of these evaluation index data. As for the page smudging problem, it can be seen from the experimental results that the model effect using cGAN is better than the other four methods. For the experiments on handwriting fading, models specializing in antique documents, such as DP-LinkNet and DocEnTr, work better on the corresponding datasets. These two model structures are able to extract deeper semantic information. For the problem of uneven illumination, it is found by the results that the methods with better processing effects are those with more a priori conditions on the dataset, such as ST-CGAN and BEDSR-Net models, which use a dataset that requires three parts: shaded image, corresponding unshaded image, and shaded mask. The different datasets targeted by BEDSR-Net can also have better results. For the deblurring problem, most current algorithms are based on GAN structure changes to encode the blurred information accurately and achieve high-quality deblurring of images. In recent years, some Transformer-based models have emerged to be applied to the deblurring problem, which handles the balance between spatial texture details and high-level context relatively well, but still does not work well for aggregating information about local and non-local pixel interactions, allowing only part of the valuable information to be passed further through the network hierarchy, and not well for passing texture details between channels. The model performance still needs to be improved. Furthermore, there are still possibilities for future development discussions.

### 4.4. Problems and Development Direction

Document image enhancement tasks are far from being solved; some tasks have only been studied minimally. In Table 5, we discuss these issues and future work.

**Table 5.** Degradation status problems or future developments.

| Degradation | Problems or Future Developments |
|---|---|
| background texture | The challenge of the background texture problem is mostly for color document images, and much of the current research is still focused on the processing of grayscale text images with better results. The color document images, on the other hand, cause a loss of text content when processed by model algorithms due to the difference in color. Therefore, methods for color document images need to be investigated. The text images include handwritten images and printed images, and there is still less discussion on the comparison of these two types of datasets, which can be processed and studied. |
| page smudge | Character recognition is very challenging when the image document has ghosting problems (ink is visible on the other side of the text page, but the ink is not precisely from the other side, causing interference with the text on the target page) and when the RGB image has a variety of ink colors and intensities that vary from moment to moment. Current methods are less effective for low-contrast document image bleed-through problems. Therefore, we need to propose a model approach based on such problems to solve these problems. |
| handwriting fading | Although the fading of handwriting will affect the readability of the document and the OCR effect, there are severe and minor fading cases, and minor fading is still relatively easy to identify. This type of degradation problem deals more with handwritten text images. For severely faded documents, the datasets of many studies are not published, and there is still a need to produce datasets for such problems. |
| the poor lighting conditions | The problem of overexposure in uneven illumination conditions, i.e., the result of adding too much flash to devices such as cameras during digitization, has been less studied [54,55]. There is no suitable dataset for this problem. Therefore, the exposure processing problem in image document enhancement still needs sufficient research in the future. |
| watermark removal | There is less research on watermark removal for document images, mainly due to the differences in language and text, resulting in limited watermark removal techniques for document images. For example, English text, a combination of 26 letters, and Chinese characters are three-dimensional homogeneous characters, resulting in less effective information redundancy in Chinese characters. Additionally, most of the methods are now only applicable to black text, the watermark color of a single document. However, watermark removal involves the integrity of the original content and intellectual property issues, so we need to handle it carefully. |
| deblurring | Models mainly generate blurred text datasets, but this is not enough to be used in real scenarios, which are diverse and many situations are not considered. The current number of blurred text datasets is still insufficient, especially since Chinese datasets are even more scarce. The insufficient coverage of scenarios in the dataset also leads to algorithmic models that do not take advantage of them. |

## 5. Conclusions

In recent years, many network models have been proposed to solve various degradation problems in document image enhancement. This article discusses six kinds of image enhancement classification problems, such as shadow removal under uneven illumination, BEDSR-Net, CANet, etc., that are proposed to solve this problem. For example, Deblur-GAN, Blur2sharp, etc., are also offered to solve the image recovery problem for blurred text. These methods can handle specific types of degradation problems very well. However, document image enhancement tasks still have some issues. For example, because of the small number of open-source datasets, many methods use GAN models to generate data, which leads to the generation of data that need to be sufficiently considered. It is not enough to be used in a wide variety of factual scenarios, and even though the models are effective for such data, the generalization ability of the models is not sufficient. Many of the current methods are based on machine learning methods for processing, but machine learning methods require data to be trained for learning. Therefore, sufficient datasets are also needed. Sufficient datasets can better refine the data and optimize the results. In the future, models trained with suitable datasets will be better able to handle document image enhancement problems.

**Author Contributions:** Conceptualization, Y.Z. and R.Z.; methodology, Y.Z.; software, Z.Y.; validation, Y.Z., J.H. and J.S.; formal analysis, Y.Z.; investigation, Z.Y.; resources, J.H.; data curation, J.S.; writing—original draft preparation, Y.Z.; writing—review and editing, Y.Z.; visualization, R.Z.; supervision, S.Z.; project administration, S.Z.; funding acquisition, S.Z. All authors have read and agreed to the published version of the manuscript.

**Funding:** This research was funded by Natural Science Foundation of Fujian Province grant number 2020J01295. And The APC was funded by Natural Science Foundation of Fujian Province.

**Institutional Review Board Statement:** Not applicable.

**Informed Consent Statement:** Not applicable.

**Data Availability Statement:** Data is contained within the article.

**Conflicts of Interest:** The authors declare no conflict of interest.

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
