# Peer review of "A Review of Document Image Enhancement Based on Document Degradation Problem"

_applsci, doi:10.3390/app13137855_

Round 1
Reviewer 1 Report
1. Why do the authors choose only six tasks of document degradation? You could have chosen more also. Is there is fixed constraint on choosing these tasks only?
2. Title should be revised, and the work six tasks must be removed.
3. Fig 2(a) is unclear. The authors should present more information on it.
4. Section 2.5 directly presents the watermark removal process. How will the reader understand about watermarking background while reading this paper? Therefore, I suggest offering some background detail on watermarking in this section by considering the recent works: https://doi.org/10.1007/s10044-022-01104-0 and https://doi.org/10.1007/s42979-022-01657-1
5. The texts and paragraphs under section 3.1 should be corrected.
6. Fig.3 needs to be enlarged for better clarity.
7. Section 3.2 should uniformly separate the paragraphs. It seems somewhere authors kept fewer contents in a paragraph and other places too much.
8. Figure 12 has low visual clarity.
9. How many images the dataset contains should be mentioned.
It is good.
Author Response
Response
Point 1: Why do the authors choose only six tasks of document degradation? You could have chosen more also. Is there is fixed constraint on choosing these tasks only?
Response 1: There is no fixed constraint, and these 6 are by far the most important tasks regarding document enhancement, many models are not specifically stated, just for degraded historical manuscript processing. The examples of the models are not very specific if more are chosen. (in red)
Point 2: Title should be revised, and the work six tasks must be removed.
Response 2: Modified on title.(in red)
Point 3: Fig 2(a) is unclear. The authors should present more information on it.
Response 3: The image has been processed.
Point 4: Section 2.5 directly presents the watermark removal process. How will the reader understand about watermarking background while reading this paper? Therefore, I suggest offering some background detail on watermarking in this section by considering the recent works: https://doi.org/10.1007/s10044-022-01104-0 and https://doi.org/10.1007/s42979-022-01657-1
Response 4: Thank you for your constructive comments. After reading the information you gave, the content in section 2.5 has been added to the background of watermark removal.(in red)
Point 5:The texts and paragraphs under section 3.1 should be corrected.
Response 5: Modified.(in red)
Point 6: Fig.3 needs to be enlarged for better clarity.
Response 6: Modified.
Point 7: Section 3.2 should uniformly separate the paragraphs. It seems somewhere authors kept fewer contents in a paragraph and other places too much.
Response 7: Thank you for your care, this paragraph has been revised to make it easier to read.(in red)
Point 8: Figure 12 has low visual clarity.
Response 8: Modified.
Point 9: How many images the dataset contains should be mentioned.
Response 9: Number of images included in the supplemented dataset.(in red)

Reviewer 2 Report
The manuscript provides a review of six known challenges that arise in the document degradation. Authors focus on deep learning methods of solving these problems, datasets, and metrics.
A wide range of papers are considered in this work (Section 3), dedicated to main tasks for image enhancement of degraded texts. However, in the number of cases, the review of existing methods is led only to their enumeration without any analysis of the features and disadvantages of models. For example, on page 6, strings 182-186, strings 191-195, strings 195-202 include only method description without their analysis.
The description of the performed experiments (Section 4.3) is not sufficient. Method of models evaluation is not described on proposed datasets. It is not clear whether the authors conduct independently training of considered models or use models with pretrained weights. It is not presented which models are taken for evaluation and whether the cross-validation stage is carried out. There are no errors during accuracy evaluation. Based on results of the review, it is difficult to form an understanding of the state-of-the art level of accuracy of the considered problems.
The conclusion (Section 5) seems incomplete. Only part of the work is described. It is claimed that some presented methods are effective to solve more than one problem. However, the experiments carried out do not contain comparisons of effectivity of one method for different problems.
The manuscript contains a number of design disadvantages:
In Introduction (page 2 strings 44-49) the structure of the work is not detailed. In this work contains 5 sections, but only 2 sections are described.
A number of figures do not have descriptions of the depicted schemes on them. It is not clear on what basis the methods for which the figures are presented were chosen.
Section 4.2 provides metric equations. For pseudo-F-Measure (page 15 string 459), the equation is claimed to be the same as F-Measure (page 14 string 451). However, Precision and Recall are not presented and differences in these metrics for both cases are not analyzed. In text of review for equation (5), there is no explanation for MAXI. Also, for equation (8), symbols are not explained. For the last described metric (page 16 strings 486-492), an equation is not presented.
On page 17 string 517, numeration of subsection is incorrect.
In this form, the manuscript is not ready for publication and significantly needs to be rewritten.
Moderate editing of English language is necessary.
Author Response
Point 1: A wide range of papers are considered in this work (Section 3), dedicated to main tasks for image enhancement of degraded texts. However, in the number of cases, the review of existing methods is led only to their enumeration without any analysis of the features and disadvantages of models. For example, on page 6, strings 182-186, strings 191-195, strings 195-202 include only method description without their analysis.
The description of the performed experiments (Section 4.3) is not sufficient. Method of models evaluation is not described on proposed datasets. It is not clear whether the authors conduct independently training of considered models or use models with pretrained weights. It is not presented which models are taken for evaluation and whether the cross-validation stage is carried out. There are no errors during accuracy evaluation. Based on results of the review, it is difficult to form an understanding of the state-of-the art level of accuracy of the considered problems.
The conclusion (Section 5) seems incomplete. Only part of the work is described. It is claimed that some presented methods are effective to solve more than one problem. However, the experiments carried out do not contain comparisons of effectivity of one method for different problems.
Response 1: Thank you very much for your constructive comments. It plays an important role in the revision of our manuscript. We have added and revised the content of the newly submitted manuscript in response to your comments about insufficient model description analysis, inadequate experimental analysis, and incomplete conclusions.(in red)
Point 2: The manuscript contains a number of design disadvantages:
In Introduction (page 2 strings 44-49) the structure of the work is not detailed. In this work contains 5 sections, but only 2 sections are described.
A number of figures do not have descriptions of the depicted schemes on them. It is not clear on what basis the methods for which the figures are presented were chosen.
Section 4.2 provides metric equations. For pseudo-F-Measure (page 15 string 459), the equation is claimed to be the same as F-Measure (page 14 string 451). However, Precision and Recall are not presented and differences in these metrics for both cases are not analyzed. In text of review for equation (5), there is no explanation for MAXI. Also, for equation (8), symbols are not explained. For the last described metric (page 16 strings 486-492), an equation is not presented.
On page 17 string 517, numeration of subsection is incorrect.
In this form, the manuscript is not ready for publication and significantly needs to be rewritten.
Response 2: To address the shortcomings in the design of the manuscript, we have clarified the structure of the work in the introduction and added and modified the content of the formula in the indicators in section 4.2.. (in red)

Reviewer 3 Report
1. The introduction and related works section is short, while the contributor is recommended to mention the proposed work's aim, goal, and research contribution. The method and implementation are not organized. There are a lot of terms and abbreviations in the paper. It would be better to have them explained a little.
2. In the result section, the training and testing information about the dataset is not mentioned in the article. What is the purpose of evaluating speed in Table 1? Only classification accuracy alone, compared, is that enough to justify the result.
3. The conclusion section is more generic and does not appear to include any analysis conclusions drawn from the evaluation data. Increase the level of depth in your description of the evaluation findings from your study approach.
4. The gap identified in the literature, rectification measures in the proposed method, and justification are unclear. Mention in the section.
5. Need to check the references. The reference format should be improved, and at the same time, many important recent references need to be updated. Check the reference numbers, and they are not in order.
Author Response
Point 1: The introduction and related works section is short, while the contributor is recommended to mention the proposed work's aim, goal, and research contribution. The method and implementation are not organized. There are a lot of terms and abbreviations in the paper. It would be better to have them explained a little.
Response 1: Thank you very much for your constructive comments. In response to the short introduction and related work section, we have added content as well as mentioned the purpose, objectives, and research contributions of the work in part 1, based on your suggestions. We also explain some terms in the text, which are reflected in part 3, such as SSP, HDC, LMD, etc. mentioned in the text.(in red)
Point 2: In the result section, the training and testing information about the dataset is not mentioned in the article. What is the purpose of evaluating speed in Table 1? Only classification accuracy alone, compared, is that enough to justify the result.
Response 2: The training and test sets of the dataset are described in the 4.1 dataset section. Not sure about the evaluation speed you are talking about in Table 1, which is a description of the number of datasets and whether they are real images or synthetic images.(in red)
Point 3: The conclusion section is more generic and does not appear to include any analysis conclusions drawn from the evaluation data. Increase the level of depth in your description of the evaluation findings from your study approach.
Response 3: The description of the conclusion was modified.(in red)
Point 4: The gap identified in the literature, rectification measures in the proposed method, and justification are unclear. Mention in the section.
Response 4: The content of the article, such as the analysis of the model and the analysis of the experimental results, was revised.(in red)
Point 5: Need to check the references. The reference format should be improved, and at the same time, many important recent references need to be updated. Check the reference numbers, and they are not in order.
Response 5: We checked the reference formatting and found errors in the formatting, such as in ref. 28, and for the rest we used the official formatting. Thank you very much for your carefulness!

Reviewer 4 Report
This survey paper is about document image enhancement, which is a pre-processing step for document analysis and recognition. The paper discusses various degradation problems that document images can face, such as smudges, fading, poor lighting conditions, and camera distortion noise. It also explores different methods to solve these problems using deep learning and convolutional neural networks. The paper reviews related work and presents common public datasets and metrics while discussing the current limitations and challenges of each degradation task. Finally, the paper provides future prospects for document image enhancement.The strength of this paper lies in its comprehensive review of various degradation problems that document images can face and the exploration of different methods to solve these problems using deep learning and convolutional neural networks. The paper also provides a detailed discussion of related work, common public datasets, and metrics for evaluating the performance of document image enhancement methods. Additionally, the paper highlights the current limitations and challenges of each degradation task and provides future prospects for document image enhancement. While this paper provides a comprehensive review of various degradation problems and methods to solve them, there are some areas that could be further explored. For example, the paper could have discussed the impact of different types of document images (e.g., handwritten vs. printed) on the performance of document image enhancement methods. Additionally, the paper could have explored the impact of different types of deep learning architectures on the performance of these methods. Finally, while the paper discusses common public datasets and metrics for evaluating document image enhancement methods, it does not provide a detailed comparison of these datasets and metrics or their limitations. Section 3.4 also serves as "problem statement" role rather "development direction". Please consider elaborate more on the proposed future work with more evidence and discussion instead of simple one sentence conclusion.
Author Response
Point 1: For your question "The paper could have discussed the impact of different types of document images (e.g., handwritten vs. printed images) on the performance of document image enhancement methods."
Response 1: Thank you very much for your constructive comments. Our dataset is including both handwritten and printed images, but we have not discussed this point in depth because the amount of zone fraction for our experiments is still not sufficient. However, in the content revision, we have some brief discussion on this in the direction of development section.(in red)
Point 2: "Explore the impact of different types of deep learning architectures on the performance of these methods."
Response 2: During the revision of the article we provide a deeper analysis of different types of deep learning architectures, especially the advantages and disadvantages of these models, and a more detailed description of the role and impact of these models on the degradation problem.(in red)
Point 3: Finally, while the paper discusses common public datasets and metrics for evaluating document image enhancement methods, it does not provide a detailed comparison of these datasets and metrics or their limitations.
Response 3: We clarify the images contained in the dataset once again, as well as add them in the text. For the metrics, we re-stated the original descriptions that were not clear enough and added the formulas.(in red)
Point 4: Section 3.4 also serves as "problem statement" role rather "development direction". Please consider elaborate more on the proposed future work with more evidence and discussion instead of simple one sentence conclusion.
Response 4: Our content describes much of the current problems of various degradation situations, and now we have added to this section with a discussion. And the title of the subsection has been modified.(in red)

Round 2
Reviewer 1 Report
The revised work has not improved as per given suggestions
Minor editing of English language required
Reviewer 2 Report
Subject to corrections, the article is ready for publication in present form.
Reviewer 3 Report
The authors responded to each and every comment by providing a suitable explanation.
There are a few typos, so the authors should check with someone who is fluent in the native language.